# World Guidance: World Modeling in Condition Space for Action Generation

Yue Su [1 2]  Sijin Chen [1]  Haixin Shi [2]  Mingyu Liu [2]  Zhengshen Zhang [2]  Ningyuan Huang [2]
Weiheng Zhong [2]  Zhengbang Zhu [2]  Yuxiao Liu [2]  Xihui Liu [1]

## Abstract

Leveraging future observation modeling to facilitate action generation presents a promising avenue for enhancing the capabilities of Vision-Language-Action (VLA) models. However, existing approaches struggle to strike a balance between maintaining efficient, predictable future representations and preserving sufficient fine-grained information to guide precise action generation. To address this limitation, we propose **WoG** (World Guidance), a framework that maps future observations into compact conditions by injecting them into the action inference pipeline. The VLA is then trained to simultaneously predict these compressed conditions alongside future actions, thereby achieving effective world modeling within the condition space for action inference. We demonstrate that modeling and predicting this condition space not only facilitates fine-grained action generation but also exhibits superior generalization capabilities. Moreover, it learns effectively from substantial human manipulation videos. Extensive experiments across both simulation and real-world environments validate that **WoG** significantly outperforms existing methods based on future prediction. Project page is available at: https://selen-suyue.github.io/WoGNet/.

## 1. Introduction

As systems designed for future action prediction, Vision-Language-Action (VLA) models (Kim et al., 2024; Black et al., 2025b;a) have been expected to improve task performance by developing a more comprehensive ability of modeling the future (Zhu et al., 2025). Recent works have

---

[1]The University of Hong Kong [2]ByteDance Seed. Correspondence to: Yuxiao Liu <liuyuxiao.876@bytedance.com>, Xihui Liu <xihuiliu@eee.hku.hk>.

*Proceedings of the 43rd International Conference on Machine Learning*, Seoul, South Korea. PMLR 306, 2026. Copyright 2026 by the author(s).

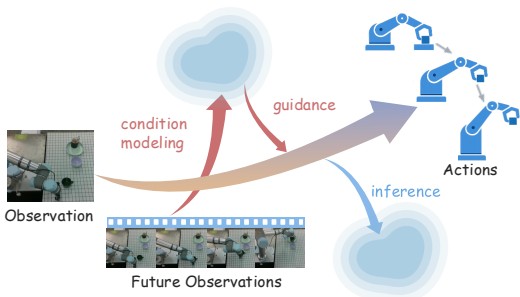

*Figure 1.* WoG first incorporates future observations into the action inference pipeline, projecting them into the condition space for action generation. Subsequently, it decouples future observations from the pipeline and simultaneously predicts these future conditions alongside actions, thereby transferring the knowledge of future conditions into the VLA model.

gone beyond predicting actions alone, investigating whether VLA models can forecast future signals in other modalities and how these predictions can be leveraged to enhance action generation (Liao et al., 2025; Cen et al., 2025b).

Existing methodologies within this landscape can be broadly categorized into two streams. (1) World Action Models (Hu et al., 2025; Cen et al., 2025b; Zhang et al., 2025c) predict explicit future modalities (such as depth, images, videos), or semantic features from foundation vision models (Oquab et al., 2024; Kirillov et al., 2023) to facilitate efficient action generation. Despite providing rich perceptual cues regarding dynamics, motion, and spatial geometry, prior work (Bu et al., 2025; Zhang et al., 2025a; Liu et al., 2025a) indicates that these generic while task-agnostic semantic spaces often contain substantial redundancy for downstream manipulation tasks. This redundancy impedes pretraining efficiency for fine-grained generation and limits cross-scenario scalability, thereby constraining their real-world performance (Bu et al., 2025; Lin et al., 2025). (2) Latent Action Models (Ye et al., 2024; Chen et al., 2025c; Bu et al., 2025) compress future actions or dynamics into sparse latent representations via reconstruction-based vision supervision, aiming to distill embodiment-agnostic high-level motion patterns. While effective for high-level planning and learnable from large-scale video data, these representations have been shown to offer only coarse guidance, lacking the precision required for fine-grained action generation (Zhang et al., 2025a; Bi et al., 2025).

Collectively, these observations underscore a fundamental trade-off. Predicting rich, task-agnostic future representations incurs significant redundancy, thereby increasing computational overhead and hampering performance (Hafner et al., 2019; Schmidt & Jiang, 2024). Conversely, compact latent action spaces typically capture only coarse motion trends, proving insufficient for fine-grained control (Bu et al., 2025; Zhang et al., 2025a). The pivotal challenge, therefore, lies in identifying a predictive space (Assran et al., 2023) that is both tractable for VLA models to forecast and sufficiently expressive to guide accurate action generation.

To address this challenge, we propose **WoG** (World Guidance), a predictive framework that operates in the condition space for action generation. We posit that to identify a *non-redundant* predictive space for world action model, the space must satisfy the criterion that its information serves as a *sufficient and effective* condition for action generation. By virtue of this role, such a space is intrinsically highly relevant to action; consequently, for a VLA model inherently designed to model actions, inferring this space becomes a tractable task. To discover such a space, we argue that the most efficient strategy is to directly incorporate future observations as conditions into the action inference pipeline (Zheng et al., 2025). The representation encoded through this pipeline thus naturally constitutes the desired efficient condition space.

Specifically, WoG follows a two-stage training curriculum. In the first stage, we implement the aforementioned design by jointly conditioning action generation on current observations, encoded by a VLM backbone, and future observations derived from frozen foundation vision models, which are queried and compressed by a trainable encoder before being integrated into the action head. This stage jointly optimizes (i) the encoder to project future observations into this efficient, implicit action-condition space, and (ii) the VLA backbone to leverage these future-conditioned representations for precise action prediction. In the second stage, we freeze the encoder to define a stable target space. The VLA is then trained to simultaneously predict this future conditioning representation and the corresponding actions, yielding a model capable of internally anticipating and utilizing future guidance during inference as shown in Figure 1.

We validate the efficiency of WoG in facilitating fine-grained action generation and ensuring robust generalization through extensive simulation and real-world experiments, where it demonstrates substantial improvements over existing methods. Furthermore, we show that WoG can be effectively improved by learning to model and predict future conditions from large-scale human videos (spanning both action-annotated and unannotated data) or UMI data (Chi et al., 2024), leading to significant performance gains in real-world robotic deployments. Our contributions are sum-

marized as follows:

- We propose a novel paradigm for world action modeling, which efficiently predicts compact future representations by modeling future observations into the condition space, thereby facilitating action generation.
- Our method demonstrates the ability to consistently model and predict future conditions in dynamic interactions, thereby supporting fine-grained action generation while maintaining strong generalization ability.
- We show that our method allows for scalable learning from a broad spectrum of data (human videos, UMI) to enhance performance and generalization ability.

## 2. Related Work

### 2.1. World Action Models

Leveraging future observation prediction (Xu et al., 2024; Su et al., 2025a; Chen et al., 2025a) to extract dynamics (Yang et al., 2025b; Hu et al., 2025) for robotic manipulation or guiding action diffusion (Sridhar et al., 2023) has been extensively studied in imitation learning (Chi et al., 2023; Zhao et al., 2023; Su et al., 2025b;c). With the recent advances in VLA models (Kim et al., 2024; Cheang et al., 2025; Black et al., 2025a; Amin et al., 2025), a growing body of work has begun to exploit the strong reasoning capabilities of Vision-Language Models (VLMs) (Karamcheti et al., 2024; Beyer et al., 2024; Yang et al., 2025a) and video generation models (Blattmann et al., 2023; Wang et al., 2025a) to perform dynamic modeling, thereby more effectively supporting action inference.

Specifically, certain approaches (Hu et al., 2025; Liao et al., 2025) introduce the intermediate features of video generation models (Blattmann et al., 2023; Agarwal et al., 2025) as world representations, integrating them into action modules to capture future manipulation dynamics. Alternative strategies (Zhang et al., 2025b; Cen et al., 2025b;a) build directly upon VLM backbones, modeling such dynamics through the internal generation of future images (Esser et al., 2020). Beyond images generation, the prediction of other explicit modalities, such as depth or optical flow, is also often incorporated into VLA co-training objectives to serve a comparable predictive role (Zhong et al., 2025; Zhang et al., 2025c). Most recently, studies have also explored directly regressing the latent representations output by foundation vision models, yielding discriminative future features that further refine the precision of action prediction (Zhang et al., 2025c). Our method stands out by learning a condition space optimized for action generation, designed to provide more efficient support for enhancing model performance.

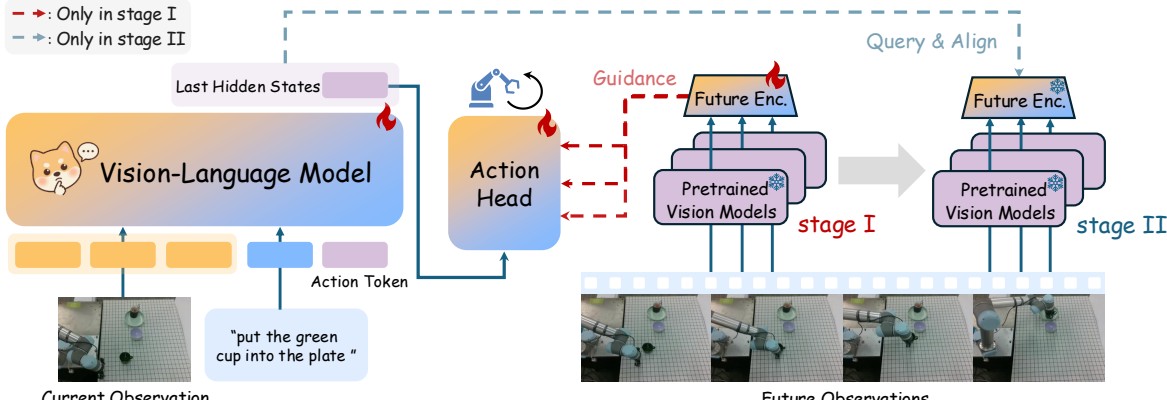

*Figure 2.* **Overview of WoG**. WoG is trained in two stages. In the first stage, future observations encoded by frozen vision foundation models are queried and compressed by a trainable Q-former-based encoder to form condition representations, which, together with VLM-encoded current observations and instructions, are used for action prediction. In the second stage, the encoder and vision models are frozen, and the VLM backbone is trained to align with the conditions while predicting actions.

## 2.2. Latent Action Models

Latent Action Models (LAMs) (Ye et al., 2024) have emerged as a response to training VLA models effectively from large-scale, heterogeneous datasets (O'Neill et al., 2024). They are built upon the assumption that, despite embodiment variations (Zhang et al., 2024), actions admit high-level representations that are skill-related but embodiment-agnostic. Such representations are typically discrete (Van Den Oord et al., 2017) and are intended to capture coarse motion trends, thereby facilitating high-level planning.

Mainstream LAMs (Ye et al., 2024; Chen et al., 2025c; Bu et al., 2025) compress heterogeneous actions across embodiments by using visual reconstruction objectives, after which a VLA model first predicts discretized latent actions and subsequently decodes them into fine-grained actions suitable for downstream tasks (AgiBot-World-Contributors, 2025). These reconstructions may rely on either generative (Ye et al., 2024) or discriminative representations (Bu et al., 2025), but are all designed to produce compact and accurate latent action spaces.

However, prior studies (Zhang et al., 2025a; 2026) have observed that the compression performed by LAMs often resembles PCA-like (Anderson, 1963) extraction of maximum-variance signals, yielding coarse planning representations and, in some cases, introducing noise from scenarios that is weakly correlated with actions. To mitigate these limitations, recent work has incorporated action reconstruction into the generative objectives of LAMs to strengthen the mapping between latent actions and control (Ma et al., 2025; Fan et al., 2026; Zhang et al., 2026). Other approaches further introduce video generation as an auxiliary co-training objective in the LAM framework, aiming to compensate for the lack of fine-grained action guidance (Ma et al., 2025; Routray et al., 2025). While sharing the goal of providing rich guidance, we posit that introducing condition prediction instead of video reconstruction in latent space is efficient.

## 3. Method

### 3.1. Problem Formulation

We consider the problem of predicting future $T$ actions $\boldsymbol{A}_{t:t+T}$ given the current observation $\boldsymbol{O}_t$ at time step $t$, and a language instruction $\boldsymbol{l}$. The VLM backbone encodes the observation and instruction as $\boldsymbol{z} \leftarrow f(\boldsymbol{O}_t, \boldsymbol{l})$, which is then fed into an action head to generate actions by maximizing the likelihood of $\boldsymbol{P}(\boldsymbol{A}_{t:t+T} \mid \boldsymbol{z})$.

In the first stage of training WoG, observations from the next $T$ future time steps are guided to be compressed into a condition space, denoted as $\boldsymbol{O}^c_{t:t+T}$. Together with the current latent representation $\boldsymbol{z}$, the compressed future condition is fed into the action head *as a guidance* to generate actions by modeling $\boldsymbol{P}(\boldsymbol{A}_{t:t+T} \mid \boldsymbol{z}, \boldsymbol{O}^c_{t:t+T})$.

Through this process, the VLA model learns to encode conditions from the current observation while leveraging complementary conditions derived from future observations for action prediction, and simultaneously acquires a compact representation of future observations.

However, given our goal of performing action inference solely based on the current observation at test time, the subset of conditions derived from the future is not accessible and must be inferred from the current observation. Under the assumption of deterministic environmental dynamics, the complete inference process should be formulated as:

$$\boldsymbol{P}(\boldsymbol{A}_{t:t+T}, \boldsymbol{O}^c_{t:t+T} \mid \boldsymbol{z})$$
$$= \boldsymbol{P}(\boldsymbol{A}_{t:t+T} \mid \boldsymbol{z}, \boldsymbol{O}^c_{t:t+T})\boldsymbol{P}(\boldsymbol{O}^c_{t:t+T} \mid \boldsymbol{z}).$$

Therefore, in the second stage, WoG is trained with two objectives. At the output of the VLM, supervision is applied

to predict the future condition by modeling $P(O^c_{t:t+T} \mid z)$. Then at the output of the action head, supervision is imposed on action prediction as $P(A_{t:t+T} \mid z)$, by optimizing this marginal action likelihood as the joint distribution under the deterministic coupling of dynamics. Together, these objectives transfer the knowledge of future condition into the VLM backbone itself. The overall process of WoG is shown in Figure 2.

### 3.2. Stage I: World Guidance

In the first stage, we use a Prismatic VLM (Karamcheti et al., 2024) adopted in OpenVLA (Kim et al., 2024) as the VLM backbone. The current observation and language instruction are encoded by the VLM backbone to obtain the latent representation $z$ (Following (Li et al., 2024a), we represent it as the output feature of the last learnable token), which is then fed into a DiT (Peebles & Xie, 2023) action head for action generation.

For future observations, they are first encoded using a combination of frozen pretrained vision models to obtain high-level representations. By default, we extract discriminant and semantic features by DINOv2 (Oquab et al., 2024) and generative features by Wan VAE Encoder (Wang et al., 2025a). Such a combination of foundation vision models is extensible and can be replaced by other pretrained visual encoders (Kirillov et al., 2023; Zhai et al., 2023). After being projected to a unified embedding dimension, these future dynamic features are processed by a learnable Q-Former-based Encoder (Li et al., 2023), which queries action-relevant features and projects them into low-dimensional conditioning representations denoted as $O^c$.

The queried representations $O^c$ are then injected into each DiT block, where they perform cross-attention with $z$ to enable action prediction conditioned on both the current and future information. We adopt rectified flow (Liu et al., 2022) to predict the velocity field:

$$\mathcal{L}_I = \mathbb{E}_{\tau, A}\left[\left\|v_\theta(A_\tau, \tau, z, O^c) - v^*\right\|_2^2\right], \qquad (1)$$

where $\tau \in [0, 1]$ denotes the scheduling timestep, $v_\theta$ denotes the predicted velocity field and $v^*$ is the target velocity.

### 3.3. Stage II: World Inference

In the second stage, the Q-Former and the projectors used for encoding future observations are frozen. WoG is designed to jointly train action prediction $P(A_{t:t+T} \mid z)$ and future condition prediction $P(O^c_{t:t+T} \mid z)$.

Specifically, we introduce learnable query embeddings to attend to the last hidden states of the VLM output with a cross-attention paradigm, and align the queried representations with the frozen future condition representations $O^c$

produced by the Encoder. Afterwards, only the VLM output $z$ is fed into the DiT head as input to predict actions. The loss is formulated as:

$$\mathcal{L}_{II} = \mathbb{E}_{\tau, A}\left[\left\|v_\theta(A_\tau, \tau, z) - v^*\right\|_2^2\right] + 1 - \mathcal{S}\left[O^c, f_q(O, l)\right], \qquad (2)$$

where $f_q(O, l)$ denotes the queried last hidden state of the VLM, and $\mathcal{S}[\cdot, \cdot]$ denotes the cosine similarity.

In this stage, the future condition is decoupled from the action head and becomes one of the prediction targets of the VLM. Through supervision, the VLM is encouraged to encode future condition information in its internal representations, enabling the model to perform complete action inference solely based on $z$. As a result, WoG is transformed into a *self-guided* model. More details about the query mechanism of WoG are provided in Appendix A.

### 3.4. Learning From Human Manipulation

Our method is also easy to extend to learn from human manipulation videos, which can be incorporated in two complementary ways.

**(1)** A small amount of human videos with action annotations is introduced in the first stage, together with robot data, to expand the condition space and capture manipulation knowledge absent from robot demonstrations. In the second stage, a much larger collection of unlabeled human videos is incorporated to supervise future condition prediction, while action supervision is applied only to robot data and optionally to the annotated human subset. This strategy enables the model to learn more sufficient and generalizable future conditions from large-scale video data.

**(2)** No action-annotated human data is required. Human videos are directly introduced in the second stage to supervise condition prediction, while the action prediction branch of human videos is masked. This setting assumes that the first stage trained on robot data already learns a sufficiently expressive condition space, and that many of these conditions, such as object motion dynamics, are shared with human manipulation videos. Under this assumption, second-stage training alone can further improve the model's ability to predict future conditions and enhance generalization across diverse scenarios.

We provide comprehensive experimental validation of both strategies in Section 5.5, demonstrating their effectiveness.

In addition, UMI can also be regarded as a universal interface for robot data. We expect that training on UMI data can further benefit our study, as it provides a means to evaluate whether the condition space of WoG can be stably modeled under egocentric observations and unseen embodiments, and subsequently transferred to our target workspace. We conduct corresponding validation in Section 5.6.

*Table 1.* **SimplerEnv evaluation across different models on Google Robot tasks.** Mv Near: Move Near, Drawer: Open/Close Drawer.

| Model | Visual Matching | | | | Variant Aggregation | | | | Overall |
| --- | --- | --- | --- | --- | --- | --- | --- | --- | --- |
| | Pick Coke | Mv Near | Drawer | Avg. | Pick Coke | Mv Near | Drawer | Avg. | Avg. |
| $\pi_0$ | 72.7% | 65.3% | 38.3% | 58.8% | 75.2% | 63.7% | 25.6% | 54.8% | 56.8% |
| $\pi_0$-FAST | 75.3% | 67.5% | 42.9% | 61.9% | 77.6% | 68.2% | **31.3%** | 59.0% | 60.5% |
| OpenVLA | 16.3% | 46.2% | 35.6% | 32.7% | 54.5% | 47.7% | 17.7% | 39.8% | 33.8% |
| GR00T-N1 | 47.0% | 70.0% | 18.1% | 45.0% | 78.8% | 62.5% | 13.2% | 51.5% | 48.4% |
| Moto | 74.0% | 60.4% | 43.1% | 59.2% | – | – | – | – | – |
| VITA | 57.5% | 55.8% | 58.9% | 57.4% | – | – | – | – | – |
| DeFI | 54.2% | 60.7% | 38.6% | 51.2% | 53.9% | 58.2% | 24.0% | 45.4% | 48.3% |
| **WoG** | **89.0%** | **82.5%** | **62.5%** | **78.0%** | **87.9%** | **75.0%** | 19.3% | **60.7%** | **69.4%** |

*Table 2.* **SimplerEnv evaluation across different models on WidowX Robot tasks.**

| Model | Put Spoon on Towel | | Stack Green on Yellow | | Put Carrot on Plate | | Put Eggplant in Basket | | Overall Average | |
| --- | --- | --- | --- | --- | --- | --- | --- | --- | --- | --- |
| | Grasp Spoon | Success | Grasp G Block | Success | Grasp Carrot | Success | Grasp Eggplant | Success | Grasp Avg. | Success Avg. |
| $\pi_0$ | 45.8% | 29.1% | 25.0% | 0.0% | 50.0% | 16.7% | 91.6% | 62.5% | 40.1% | 27.1% |
| $\pi_0$-FAST | 62.5% | 29.1% | 58.5% | 21.9% | 54.0% | 10.8% | 83.3% | 66.6% | 48.3% | 32.1% |
| OpenVLA | 4.1% | 0.0% | 33.0% | 0.0% | 12.5% | 0.0% | 8.3% | 4.1% | 7.8% | 1.1% |
| GR00T-N1 | 83.3% | 62.5% | 54.2% | 45.8% | 70.8% | 16.7% | 41.7% | 20.8% | 49.5% | 36.5% |
| UniVLA | 76.4% | 52.8% | 66.7% | 2.8% | **79.2%** | **55.6%** | 87.5% | 66.7% | 77.5% | 45.6% |
| ViPRA | 79.2% | 66.7% | 62.5% | **54.2%** | 54.2% | 50.0% | 91.7% | 79.2% | 71.9% | 62.5% |
| **WoG** | **95.8%** | **79.2%** | **75.0%** | 33.0% | 70.8% | 50.0% | **100.0%** | **91.7%** | **85.4%** | **63.5%** |

# 4. Simulation Experiments

## 4.1. Setup

**Evaluation Setup.** We evaluate our method in the SIM-PLER simulation environment (Li et al., 2024b), which includes two robotic configurations: the Google Robot and WidowX. During evaluation, the model operates in a closed-loop manner and receives only a single RGB observation at each time step. Detailed training and evaluation settings of our method can be found in Appendix B.

**Baselines.** In the baseline comparison, we aim to highlight both the advantages of WoG over conventional VLA approaches and its distinctions from closely related methods. Accordingly, we include a broad and representative set of VLA baselines spanning multiple methodological paradigms. Specifically, our comparisons cover:

(i) Conventional VLA methods that directly map visual-language observations to actions: $\pi_0$ (Black et al., 2025b), $\pi_0$-FAST (Pertsch et al., 2025), OpenVLA (Kim et al., 2024) and GR00T-N1 (Bjorck et al., 2025);

(ii) Latent Action Models: Moto (Chen et al., 2025c) and UniVLA (Bu et al., 2025);

(iii) World Action Models that leverage video prediction to capture dynamics: DeFI (Anonymous, 2026);

(iv) Methods that jointly perform latent action modeling with future video generation: VITA (Ma et al., 2025) and ViPRA (Routray et al., 2025).

## 4.2. Quantitative Results

Most tasks in the SIMPLER benchmark belong to the pick-and-place type, where successful execution critically depends on both dynamic trajectory planning and precise end-effector pose prediction. In particular, obstacle avoidance during motion requires anticipating scene dynamics, while grasping and placement demand accurate reasoning about future contact and collision constraints.

As shown in Table 1 and Table 2, WoG achieves strong and consistent performance improvements over all baselines across the majority of tasks. In scenes where interfering objects are present, such as *Move Near*, our approach exhibits markedly superior trajectory planning behavior, effectively navigating dynamic interference and maintaining stable execution. This highlights the benefit of incorporating future-aware conditions for motion reasoning under complex environmental dynamics.

Furthermore, in broad P&P tasks like *Pick Coke* and *Put Spoon*, WoG substantially improves the accuracy of future grasp and placement pose prediction. By extracting future semantics in a manner that is both sufficient and compact, our approach achieves superior accuracy in estimating target poses, thereby enhancing positioning precision and collision avoidance capabilities. It is particularly noteworthy that, while our paradigm shares the similar underlying set of future observations with methods combining latent actions with future video generation, it diverges fundamentally in how this information is utilized. Unlike full-scale video prediction, which aims to reconstruct all visual information, our method selectively extracts critical semantics into the condition space. This strategy yields a more robust repre-

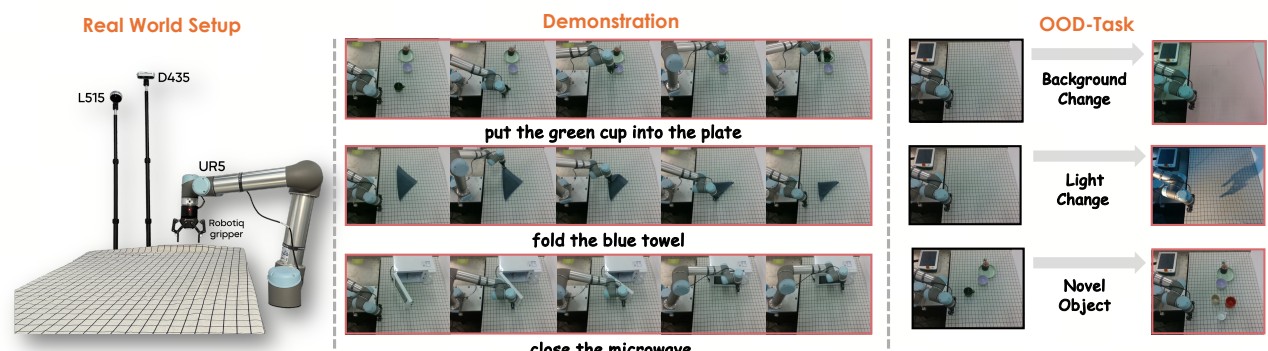

*Figure 3.* **Overview of our real-world experiment setup**. The figure shows our robotic platform and sensors (left), the execution of the three tasks under the in-distribution setup (middle), and the modifications applied for the out-of-distribution setup (right).

*Table 3.* Ablations on the condition space design in SIMPLER. "Full DINO" denotes directly aligning VLM representations with the full DINOv2 semantic space using 256 alignment tokens. "64/16/8 tok." denote WoG with different condition-space capacities using 64, 16, and 8 query tokens respectively.

| Model | Visual Matching | Variant Aggregation | Overall Avg. |
|---|---|---|---|
| Vanilla VLA | 68.4% | 50.3% | 59.4% |
| Full DINO | 71.9% | 53.7% | 62.8% |
| 64 tok. | 77.3% | 56.5% | 66.9% |
| 16 tok. | **79.3%** | **59.6%** | **69.5%** |
| 8 tok. | 73.6% | 58.3% | 66.0% |

sentation that effectively mitigates the propagation of visual prediction errors into the action space, ultimately facilitating action prediction with significantly higher precision.

We also note that in a small subset of tasks characterized by action constraints, such as *Stack Green on Yellow* and *Drawer* for their requirement on accurate relative position of stacks or gripper and drawer, the performance gains are comparatively smaller. This is primarily attributable to the limited spatial resolution of the current backbone and the inherent difficulty of modeling fine-grained geometry (Zhang et al., 2025d; Chen et al., 2025b) through current dynamic prediction alone.

**4.3. Effectiveness of the Condition Space Design**

To further validate the effectiveness of the proposed condition space, we conduct additional ablations in SIMPLER using DINOv2 as the future encoder. We compare: (1) a vanilla VLA without future condition modeling; (2) a variant that directly aligns VLM embeddings with the full DINOv2 semantic space using 256 alignment tokens; and (3) WoG with different condition-space capacities using 64, 16, and 8 query tokens respectively. The results are shown in Table 3.

Directly predicting the full semantic space only brings limited improvement over the vanilla VLA, suggesting that

dense semantic representations contain substantial redundancy for downstream action generation. In contrast, learning a compact condition space consistently improves performance.

We further observe a clear trade-off between expressiveness and generalization. Increasing the condition-space capacity to 64 tokens improves Visual Matching performance but generalizes less effectively under Variant Aggregation. Conversely, the more compact 8-token condition space achieves stronger robustness but sacrifices fine-grained action precision. Overall, the 16-token setting achieves the best balance, validating our hypothesis that an effective predictive space for action generation should remain compact while preserving sufficient action-relevant future information.

## 5. Real-World Experiments

### 5.1. Setup

**In-Distribution (ID) Setup.** We design the following 3 tasks for evaluation and collect corresponding expert demonstrations, forming our In-Distribution (ID) setup:

*Pick and Place (P&P)* is a rigid-body manipulation, where the robot picks and places a green cup into a plate following the given instruction. Successful execution requires avoiding table-top obstacles and collisions with objects already in the plate, which evaluates the model's ability to predict action-relevant dynamics for collision-aware trajectory planning. We collect 100 expert demonstrations for this task.

*Close the Microwave* is an articulated-object manipulation task, where the robot is required to close the microwave door. This task evaluates the policy's ability to predict and control articulated rotational dynamics. We collect 100 expert demonstrations for this task.

*Fold the Towel* is a deformable manipulation task, where the robot grasps one bottom corner of a towel folded into an

isosceles triangle and aligns it with the other bottom corner. The task is considered successful if the distance between the two corners is less than $5\,\mathrm{cm}$. We collect 200 expert demonstrations for this task.

**Out-of-Distribution (OOD) Setup.** To evaluate generalization, we construct 3 conditions unseen in the expert demonstrations, forming the Out-of-Distribution (OOD) setup:

*Background Change*: Expert demonstrations are collected using a single tablecloth, while evaluation is conducted with a different tablecloth to introduce background variation.

*Light Change*: During evaluation, a numerically controlled light source with fixed intensity is projected onto the workspace from a fixed position, introducing lighting conditions not covered by the expert demonstrations.

*Novel Object*: For *Fold the Towel*, evaluation is conducted using a unseen towel. For *Pick and Place*, we replace the original cup with cups of different colors and shapes and modify the corresponding instructions accordingly.

**Platform.** We use a UR5 robotic arm equipped with a Robotiq 2F-85 gripper for manipulation. A top-down Intel RealSense D435 and an L515 camera are mounted to provide global observations, while only the D435 is used in this work. All devices are connected to a workstation with an NVIDIA RTX 4090 GPU for model inference and control.

**Baselines.** We select UniVLA (Bu et al., 2025), a Latent Action Model, and VPP (Hu et al., 2025), which leverages future video prediction, both of which have been widely validated in real-world robotic settings, as baselines. In addition, we compare against variants that ablate each training stage, as well as models trained on different data sources, to comprehensively assess the contribution of our approach.

**Protocols.** For the real-world evaluation, 20 trials are performed per method for each task. All methods are evaluated under closely matched randomized initial scene configurations for each trial.

The above experiment setups are shown in the Figure 3. Detailed training and evaluation settings of our method can be found in Appendix C.

### 5.2. Effective and Generalizable Manipulation

The performance results of WoG and baselines are summarized in Table 4. Through these evaluations, we aim to assess whether WoG can effectively model future condition and leverage its prediction to facilitate action generation in dynamic interactions. We further examine whether incorporating future observation prediction leads to overfitting to ID scenarios, potentially degrading generalization performance under OOD setup.

**Fine-grained action prediction in dynamic interactions.**

*Table 4.* **Detailed performance of WoG and baselines.** The left side of the arrow is the performance of the In-Distribution setup, and the right side is performance of the Out-of-Distribution setup.

| ID setup | | UniVLA | VPP | WoG |
|---|---|---|---|---|
| **Pick and Place** | | 25% | 55% | **60%** |
| **Close the Microwave** | | 80% | 90% | **100%** |
| **Fold the Towel** | | 20% | 45% | **60%** |

| OOD setup | Task | UniVLA | VPP | WoG |
|---|---|---|---|---|
| **Background Change** | P&P | 25%→20% | 55%→30% | 60%→**55%** |
| | Fold | 20%→20% | 45%→30% | 60%→**50%** |
| **Light Change** | Fold | 20%→10% | 45%→20% | 60%→**35%** |
| **Novel Object** | P&P | 25%→10% | 55%→15% | 60%→**40%** |
| | Fold | 20%→10% | 45%→30% | 60%→**50%** |

Our tasks challenges the model to handle rigid, articulated, and deformable objects, requiring precise low-level control under complex dynamics. In the *P&P* task, which necessitates simultaneous obstacle avoidance and precise end-effector placement, UniVLA falters due to the coarse resolution of its high-level latent planning. While VPP captures dynamics via video prediction, it is consistently outperformed by WoG. This comparison underscores the advantage of explicitly modeling predictive information in a dedicated action condition space, rather than relying on high-dimensional visual predictions. This advantage is further amplified in the *Fold* task. Controlling deformable objects requires accurate trajectory planning and precise release timing to achieve the target geometry. WoG significantly widens the performance gap over VPP, as its condition space effectively distills manipulation-relevant dynamics (e.g., cloth deformation) while discarding the redundant perceptual signals inherent in video generation. Similarly, in the *Close* task, WoG achieves near-perfect performance, confirming that the predicted conditions are sufficient to guide fine-grained interaction with articulated dynamics.

**Generalization ability in OOD scenes.** A key aspect of generalization from upstream pretraining to downstream finetuning is the ability to transfer manipulation knowledge across heterogeneous embodiments, while preserving reliance on high-level visual representations rather than overfitting to the finetuning environment. Under background and novel object variations, baseline methods exhibit pronounced degradation. Latent action models tend to implicitly overfit to object-specific dynamics coupled with training appearances, limiting transferability. Similarly, VPP is constrained by the visual distribution of expert demonstrations, leading to artifacts when inputs deviate from the training domain. In contrast, WoG maintains superior performance with minimal degradation across all OOD scenarios. We attribute this robustness to the design of the condition space: by querying and compressing informative features from *frozen*, pretrained visual encoders, WoG constructs

conditions that are highly distinctive for manipulation yet invariant to visual nuisances. This design ensures that the generalization power of upstream visual priors is preserved rather than distorted during fine-tuning. Notably, even under severe lighting changes, as the most challenging shift, WoG exhibits the smallest relative performance drop. This confirms that our framework learns stable, action-centric representations, striking an optimal balance between expressiveness for control and compactness for generalization.

## 5.3. Efficient Training Strategy

*Table 5.* **Ablation of training stages in WoG.** *Vanilla VLA* predicts actions from current observations using our VLM and DiT head. *WoG w/o cotrain* applies the same first-stage training of WoG and second-stage training without condition supervision.

| ID setup | | vanilla VLA | WoG w/o cotrain | WoG |
|---|---|---|---|---|
| **Pick and Place** | | 45% | 45% | **60%** |
| **Close the Microwave** | | 90% | 95% | **100%** |
| **Fold the Towel** | | 40% | 30% | **60%** |

| OOD setup | Task | vanilla VLA | WoG w/o cotrain | WoG |
|---|---|---|---|---|
| **Background Change** | P&P | 45%→45% | 45%→45% | 60%→**55%** |
| | Fold | 40%→25% | 30%→30% | 60%→**50%** |
| **Light Change** | Fold | 40%→10% | 30%→10% | 60%→**35%** |
| **Novel Object** | P&P | 45%→**40%** | 45%→35% | 60%→**40%** |
| | Fold | 40%→30% | 30%→30% | 60%→**50%** |

We pretrain WoG and its variants on the OXE dataset under different training strategies as ablations. *vanilla VLA* follows standard VLA pretraining with the same VLM backbone and DiT head, supervising actions from current observations only. *WoG w/o cotrain* is trained using the first-stage future-observation-guided objective and second-stage action supervision without condition supervision. All models are fine-tuned with identical expert demonstrations on downstream tasks. Details are shown in Appendix C.3

As shown in Table 5, WoG trained with the two-stage scheme outperforms both variants. Compared to the *vanilla VLA*, WoG achieves improvements across all tasks under the ID setup, demonstrating its ability to extract effective conditions from observations. Moreover, under the OOD setup, WoG maintains strong generalization performance, indicating the robustness of the predictive condition space. For *WoG w/o cotrain*, we observe performance comparable to the *vanilla VLA* across tasks. This suggests that introducing future conditions does not significantly degrade the action generation capability of the VLM backbone. However, this variant still falls notably behind the full WoG, highlighting the necessity of the co-training. These results confirm that explicitly supervising the alignment between future conditions and the VLM backbone is crucial for distilling future action-relevant knowledge into the VLA model.

## 5.4. Effect of Condition Guidance on Action Prediction

To further examine whether the learned condition $O^c$ provides effective guidance for action prediction, we conduct a validation-set analysis on the real-world data. We fine-tune both the standard WoG and a Stage-I-only variant using 95% of the data, and report normalized action MSE on the remaining 5% validation set. For the Stage-I-only variant, we evaluate two inference modes: with and without access to future observations encoded as $O^c$.

*Table 6.* Action prediction error under different inference settings. The Stage-I model is evaluated with and without access to the future condition $O^c$ constructed from ground-truth future observations. Lower values are better.

| Method | P&P ↓ | Fold ↓ | Close ↓ |
|---|---|---|---|
| Stage I w/o $O^c$ | 0.043 | 0.058 | 0.018 |
| Stage I w. $O^c$ | **0.007** | **0.018** | **0.005** |
| WoG | 0.011 | 0.022 | 0.006 |

As shown in Table 6, removing $O^c$ from the Stage-I model leads to substantially larger action prediction errors, while introducing future-conditioned representations significantly improves prediction accuracy. This confirms that the learned condition space provides direct and effective guidance for action generation. Moreover, the standard WoG closely approaches the performance of the Stage-I model with ground-truth future observations, indicating that Stage-II co-training successfully transfers future-condition knowledge into the VLA backbone and enables self-guided inference without access to future frames.

## 5.5. Learning from Human Data

*Table 7.* **Detailed performance under different human data integration strategies.** *w/o human data*: the original WoG trained solely on robot data. *w. human v.*: WoG trained with human videos without action annotations. *w. human v./a.*: WoG trained with a small subset of action-annotated human videos and a larger subset of unannotated human videos.

| ID setup | | w/o human data | w. human v. | w. huamn v./a. |
|---|---|---|---|---|
| **Pick and Place** | | 60% | **70%** | **70%** |
| **Fold the Towel** | | 60% | 50% | **65%** |

| OOD setup | Task | w/o human data | w. human v. | w. huamn v./a. |
|---|---|---|---|---|
| **Background Change** | P&P | 60%→55% | 70%→**70%** | 70%→**70%** |
| | Fold | 60%→50% | 50%→45% | 65%→**60%** |
| **Light Change** | Fold | 60%→35% | 50%→30% | 65%→**45%** |
| **Novel Object** | P&P | 60%→40% | 70%→35% | 70%→**45%** |
| | Fold | 60%→**50%** | 50%→45% | 65%→50% |

We evaluate the two human data utilization strategies introduced in Section 3.4, with results summarized in Table 7. *w. human v.* denotes leveraging unannotated human videos only in the second stage for future condition supervision. *w. human v./a.* further incorporates a small subset of action-

annotated human videos, which are supervised for action prediction in both training stages. Specific dataset information and training strategies are introduced in Appendix C.4.

As shown in Table 7, we observe that even when human videos are used solely for condition prediction supervision in the second stage, our model still benefits on the P&P task and exhibits a smaller relative performance drop under OOD settings. However, performance degrades on the deformable object manipulation task. We attribute this behavior to the task-dependent similarity between human and robotic manipulation. For pick-and-place, task execution patterns in human demonstrations closely resemble robotic behaviors, leading to more aligned action-relevant conditions. In contrast, deformable object manipulation induces a larger mismatch in the condition space for many conditions from more flexible human manipulations are not modeled during first-stage robot training, which limits the transferability of human manipulation knowledge and results in degraded performance.

Nevertheless, even with only 220h of action-annotated human data introduced in the first training stage, WoG is able to rapidly acquire human-aligned conditioning representations and effectively transfer them to robotic manipulation. The resulting model consistently outperforms its robot-only counterpart across all ID and OOD settings, demonstrating substantially improved generalization. These results indicate the strong potential of our framework to scale with larger and more diverse human datasets.

### 5.6. Learning from UMI Data

We collected an additional 120 UMI trajectories for both the *P&P* and *Fold* tasks to augment the training process detailed in Appendix C.5. Crucially, this data was introduced exclusively during the second-stage fine-tuning alongside our expert demonstrations, ensuring that the condition space learned in the first stage remained unaltered. Despite the significant domain gaps characterized by UMI's egocentric observations, distinct action representations, and a completely different embodiment configuration: the fact that the condition space was established solely on OXE pretraining without prior exposure to such inputs, WoG achieved remarkable performance gains.

Specifically, success rates surged from 60% to 85% on *P&P* and from 60% to 80% on *Fold* as shown in Figure 4. These results demonstrate that even when pretrained strictly on standard robot data, WoG acquires a highly robust and generalizable condition encoding capability. We attribute this success to the model's proficiency in capturing embodiment-agnostic dynamics, such as intrinsic object motion, which facilitates the seamless integration of UMI data and highlights the broad scalability of our approach.

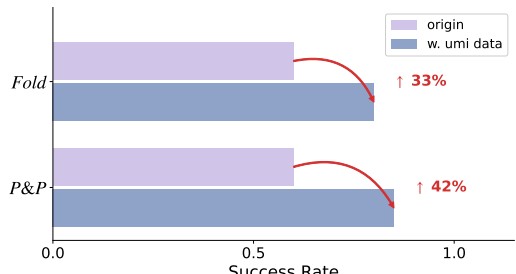

*Figure 4.* **Performance after training with UMI data.** Compared to training solely with robot data, the WoG showed a 42% improvement in performance on the P&P task and a 33% on the Fold task.

## 6. Conclusion

We propose WoG, a novel world modeling framework for action generation that compresses future observations into a low-dimensional condition space via guidance. By jointly predicting these compact future conditions within a VLA, WoG strikes an effective balance between efficient future forecasting and the acquisition of rich manipulation knowledge. Experiments demonstrate the efficiency and strong generalization capability of our approach, while validating the effectiveness of the proposed training strategy and the scalability of WoG to large-scale human manipulation data. Future work may focus on designing more expressive and efficient condition representations to better handle scenarios with strong spatial or action constraints, as well as exploring improved knowledge distillation and more generalizable condition learning from large-scale human videos.

## Acknowledgment

The research work described in this paper was conducted in the JC STEM Lab of Autonomous Intelligent Systems funded by The Hong Kong Jockey Club Charities Trust.

## Impact Statement

This work aims to advance the robotics community's perspective on the auxiliary utility of multi-modal future signals. We critically examine effective strategies for integrating information from existing generative and discriminative vision models into downstream robotic tasks. We posit that within the extensive feature spaces of foundation vision models, there exists a task-relevant *conditional subset*; identifying this subset is sufficient to facilitate highly efficient inference in VLA models. From the perspective of the broader machine learning community, the impact of this work lies in providing an efficient methodology for adapting and leveraging general-purpose foundation models for specific downstream applications.

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

# A. Detailed Architecture

Our method samples future visual observations at a temporal frequency of one-quarter relative to the action sequence. Concretely, for the default prediction horizon of 16 action steps, we uniformly sample 4 frames to serve as the future observation sequence. Regarding feature encoding, the frozen DINOv2 model processes each sampled image to extract semantic features. In parallel, for the Wan VAE, we utilize the current observation as the initiating frame, encoding it jointly with the sampled future frames to capture temporal and spatial features. Both the DINOv2 features and the VAE features (with their last two spatial dimensions flattened) are projected into a unified embedding space matching the hidden dimension of the Q-Former and are subsequently stacked. A Q-Former, instantiated with $N = 16$ learnable query tokens, aggregates the necessary dynamic representations from this combined feature set via cross-attention mechanisms, ultimately compressing the information into a compact condition space with a dimensionality of $D = 32$ by default.

The query mechanism of the Q-Former-based Encoder remains consistent across both training stages, with the sole distinction being that the encoder transitions from a trainable state in the first stage to a frozen state in the second. In this second stage, the VLM processes the current observation to yield the sequence of last hidden states. Capitalizing on the causal nature of the architecture, the trailing tokens effectively distill the most comprehensive visual and linguistic context—analogous to our strategy of utilizing the final learnable token as the action token. Consequently, we select the last 4 tokens from the VLM output to align with the target future representations extracted by the frozen vision foundation models. To implement this alignment, we instantiate 16 learnable query embeddings, matching the token count of the condition representation. These embeddings perform cross-attention over the selected VLM hidden states and are subsequently projected into the 32-dimensional space to compute the alignment loss against the ground-truth future conditions.

We detailed above query mechanisms in Figure 5.

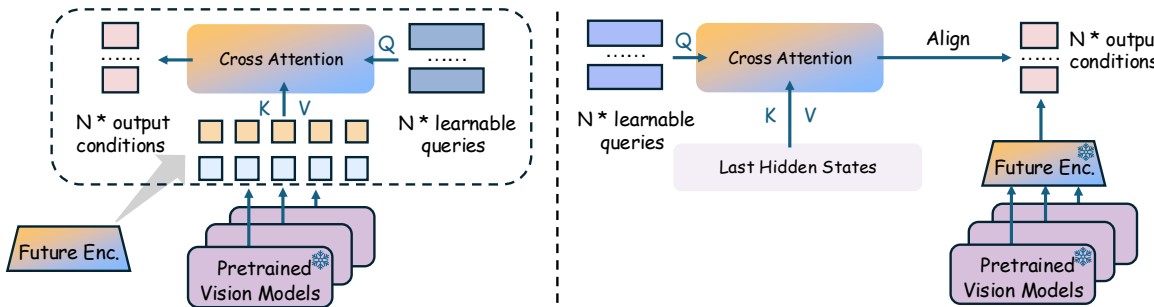

*Figure 5.* **Detailed illustration of the query mechanisms within WoG.** The **left panel** depicts the future encoder, which maintains a set of learnable queries to extract low-dimensional, action-relevant conditions from the features of pretrained vision models. The **right panel** illustrates the query mechanism for condition prediction during the second training stage. Here, a set of learnable query embeddings performs cross-attention with the last hidden states of the VLM, producing predictive representations that are supervised to align with the target conditions generated by the now-frozen future encoder.

# B. Simulation Experiments

### B.1. Training Settings

For simulation evaluation, Our model is pretrained on the Open X-Embodiment (OXE) dataset (O'Neill et al., 2024) for 100k training steps with a global batch size of 1024 in the first stage, the dataset sampling ratios are shown in Table 8. In the second stage, the model is further trained on the Bridge and Fractal datasets for 50k steps using the same batch size with individual ratios as about 51% and 49%. No additional fine-tuning is needed on individual Bridge or Fractal datasets.

### B.2. Evaluation Settings

The SIMPLER simulator (Li et al., 2024b) distinguishes itself by functioning as a rigorous *real-to-sim* evaluation proxy for the Fractal and Bridge datasets, thereby compelling models to tackle cross-domain transfer challenges. Notably, the simulator offers two evaluation protocols characterized by distinct degrees of domain shift: *Visual Matching*, which aims to closely replicate real-world tasks by minimizing visual discrepancies between simulated and physical environments;

*Table 8.* Dataset Sampling Ratios during the OXE pretraining.

| Dataset | Ratio | Dataset | Ratio | Dataset | Ratio |
|---|---|---|---|---|---|
| Fractal | 0.140639 | Kuka | 0.140674 | Bridge | 0.146649 |
| Taco Play | 0.032661 | Jaco Play | 0.005354 | Berkeley Cable Routing | 0.002907 |
| Roboturk | 0.025753 | Viola | 0.010483 | Berkeley Autolab UR5 | 0.013452 |
| Toto | 0.022367 | Stanford Hydra Dataset | 0.049202 | Austin Buds Dataset | 0.002343 |
| NYU Franka Play Dataset | 0.009245 | Furniture Bench Dataset | 0.027112 | UCSD Kitchen Dataset | 0.000545 |
| Austin Sailor Dataset | 0.024248 | Austin Sirius Dataset | 0.019224 | DLR EDAN Shared Control | 0.000613 |
| IAMLab CMU Pickup Insert | 0.010043 | UTAustin Mutex | 0.024852 | Berkeley Fanuc Manipulation | 0.008600 |
| CMU Stretch | 0.001718 | BC-Z | 0.082621 | FMB Dataset | 0.023224 |
| DobbE | 0.015656 | DROID | 0.111433 | | |

and *Variant Aggregation*, which introduces more severe distributional shifts by modifying environmental elements such as backgrounds, lighting conditions, distractors, and table textures.

In this evaluation, to assess the adaptability of our framework, we instantiate WoG with 3 different combinations of foundation vision encoders: (i) DINOv2 (Oquab et al., 2024) only, (ii) DINOv2 paired with SigLIP (Zhai et al., 2023), and (iii) DINOv2 paired with the Wan VAE encoder (Wang et al., 2025a). The results of these configurations are presented in Table 9 and Table 10.

*Table 9.* **SimplerEnv evaluation across different pretrained encoder configurations on Google Robot tasks.** Mv Near: Move Near, Drawer: Open/Close Drawer.

| Model | Visual Matching | | | | Variant Aggregation | | | | Overall |
|---|---|---|---|---|---|---|---|---|---|
| | Pick Coke | Mv Near | Drawer | Avg. | Pick Coke | Mv Near | Drawer | Avg. | Avg. |
| WoG (dino) | 96.0% | 85.8% | 56.0% | 79.3% | 88.1% | 81.0% | 9.8% | 59.6% | 69.5% |
| WoG (dino-siglip) | 89.0% | 82.5% | **62.5%** | 78.0% | 87.9% | 75.0% | **19.3%** | **60.7%** | 69.4% |
| WoG (dino-vae) | **97.7%** | **87.1%** | 61.1% | **82.0%** | **89.7%** | **77.5%** | 12.4% | 59.9% | **70.9%** |

*Table 10.* **SimplerEnv evaluation across different different pretrained encoder configurations on WidowX Robot tasks.**

| Model | Put Spoon on Towel | | Stack Green on Yellow | | Put Carrot on Plate | | Put Eggplant in Basket | | Overall Average | |
|---|---|---|---|---|---|---|---|---|---|---|
| | Grasp Spoon | Success | Grasp G Block | Success | Grasp Carrot | Success | Grasp Eggplant | Success | Grasp Avg. | Success Avg. |
| WoG (dino) | 83.3% | 66.7% | 41.7% | 8.3% | 50.0% | 29.2% | **100.0%** | 91.7% | 68.8% | 49.0% |
| WoG (dino-siglip) | **95.8%** | **79.2%** | **75.0%** | **33.0%** | 70.8% | **50.0%** | **100.0%** | 91.7% | 85.4% | **63.5%** |
| WoG (dino-vae) | **95.8%** | 54.2% | 66.7% | 29.2% | **83.3%** | **50.0%** | **100.0%** | **100.0%** | **86.4%** | 58.4% |

The quantitative results presented in Table 9 and Table 10 reveal three key insights regarding the design of the condition space:

- **Benefits of Enriched Representations:** Both the SigLIP and Wan VAE augmented configurations consistently outperform the distinct DINOv2 baseline. This validates that enriching the condition space with high-level semantics or temporal dynamics significantly enhances policy robustness.

- **VAE for Trajectory Planning:** The *dino-vae* variant demonstrates superior performance on Google Robot tasks (e.g., *Pick Coke*, *Move Near*), achieving the highest overall success rate of 70.9%. We attribute this to the VAE encoder's ability to compress spatiotemporal information, which effectively aids the policy in modeling object dynamics and planning smooth trajectories under environmental disturbances.

- **SigLIP for Spatial Precision:** On tasks that demand fine-grained spatial reasoning, such as *Stack Green on Yellow*, the *dino-siglip* variant exhibits a distinct advantage, surpassing *dino-vae* by a notable margin in success rate (33.0% vs. 29.2%). This suggests that explicit high-level semantic alignment provided by SigLIP is critical for handling tasks with spatial constraints and precise positioning requirements.

It's noteworthy that the integration of SigLIP enriches the condition space with high-level semantics, partially mitigating spatial precision deficits and ensuring robust performance across the SIMPLER benchmark. Thus, we utilize the *dino-siglip*

configuration as the default for simulation. Nevertheless, we contend that modeling fine-grained spatial constraints remains a persistent challenge independent of the future encoder. Solving this requires dedicated spatial mechanisms (Wang et al., 2025b; Zhuo et al., 2025) or historical observation modeling, which lies beyond the scope of this work. Since our real-world experiments are tailored to validate trajectory planning, we retain the VAE-incorporated framework in physical evaluations to fully exploit its superior capacity.

**Lightweight visual encoders.** We further evaluate whether WoG relies on a heavy pretrained future encoder. Specifically, we compare the default WoG using a pretrained 300M DINOv2 encoder with a variant using a lightweight 80M ViT encoder trained from scratch in Stage I. The results are summarized in Table 11.

*Table 11.* Evaluation with different future visual encoders in SIMPLER. "Light ViT" denotes an 80M ViT encoder trained from scratch in Stage I, while "DINOv2" denotes the default pretrained 300M DINOv2 encoder.

| Model | Visual Matching | Variant Aggregation | Overall Avg. |
|---|---|---|---|
| WoG (Light ViT) | 74.5% | 56.6% | 65.6% |
| WoG (DINOv2) | **79.3%** | **59.6%** | **69.5%** |

As shown in Table 11, WoG still achieves strong performance with the lightweight encoder trained from scratch, indicating that condition-space learning does not strictly depend on large pretrained visual representations. Meanwhile, the pretrained DINOv2 encoder yields stronger performance, especially under Variant Aggregation, suggesting that pretrained visual priors provide more robust representations for modeling future conditions across scene variations.

# C. Real-World Experiments

## C.1. Training Settings

For real-world experiments, our model is pretrained on the Open X-Embodiment (OXE) dataset for 100k training steps with a global batch size of 1024 in the first stage and 50k steps on OXE in the second stage. We also use the OXE pretrained checkpoints provided by the baselines. All the models are then finetuned on our real-world episodes for 30 epochs with a global batch size of 512 (During the fine-tuning phase, WoG mirrors the training protocol of only the second stage, maintaining the co-training objective that jointly optimizes for both action prediction and future condition forecasting). The oxe pretraining dataset sampling ratios are shared with Table 8. For our self-collected finetuning dataset, all the tasks share equal sampling weights, Therefore, the sampling probability is only positively correlated with the training sample amount for each task.

## C.2. Evaluation Settings

We adopt Maniunicon (Liu et al., 2025b) control strategy with the same evaluation settings. For all methods, the policy predicts a future action sequence of length 16 conditioned on the current RGB observation from the D435 camera, of which the first 8 steps are executed. As mentioned above, we continue to use the default DINOv2 and Wan VAE as the pre-trained visual encoder combination for our real-world experiments.

## C.3. Ablation Details

For the *Vanilla VLA*, we adhere to a standard training protocol where the model is supervised solely on action prediction given current observations. To ensure a fair comparison, we align the training budget with the total duration of the two-stage WoG: the model is pretrained on the OXE dataset for 150k steps with a global batch size of 1024, followed by fine-tuning on expert demonstrations using a batch size of 512 for 30 epochs.

Regarding the *WoG w/o cotrain* variant, the first stage setup remains identical to that of the full WoG, involving 100k steps of pretraining with a batch size of 1024 under future-observation-guided action supervision. In the second stage, the model continues training for an additional 50k steps with the same batch size. However, crucially, we exclude the co-training objective in this phase; the model receives supervision exclusively for action prediction, without the auxiliary supervision for predicting future conditions. Subsequently, the model undergoes fine-tuning on expert demonstrations for 30 epochs with a global batch size of 512. Consistent with the second pretraining stage, supervision during this fine-tuning phase remains restricted exclusively to action prediction.

## C.4. Human Data Learning Settings

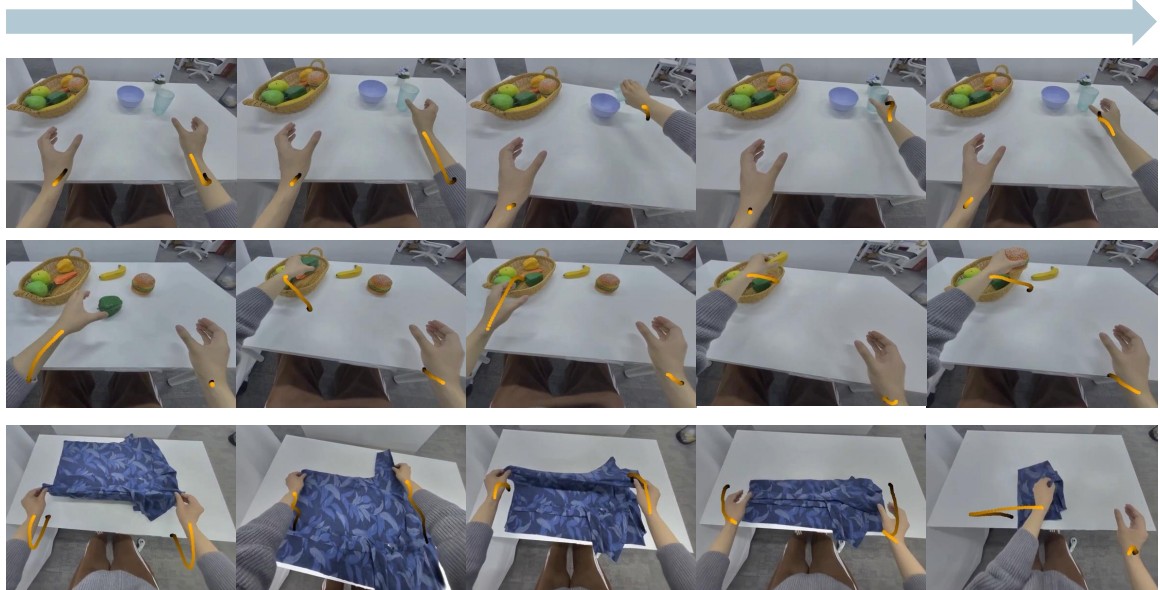

*Figure 6.* **Process of human manipulation process.** We keep the same human manipulation data collection strategy as in (Cheang et al., 2025; Wen et al., 2026). Human trajectories can be efficiently collected with VR devices at a rate of approximately 450 trajectories per hour, substantially outpacing the teleoperated robot trajectory collection. Nevertheless, as discussed earlier, we use action annotations for only 11% of the data, while the remaining 89% are leveraged solely as unlabeled videos. We argue that this setting more faithfully reflects real-world conditions, where action-labeled human manipulation data are scarce but large-scale unlabeled videos are abundant, and is therefore essential for evaluating the scalability of WoG.

**Dataset Information.** We use a human manipulation dataset collected by ourselves with PICO 4 Ultra Enterprise as in (Cheang et al., 2025), consisting of **650k** trajectories with a total duration of approximately **1,920** hours, among which a **220**-hour subset is annotated with actions. The human arm state is parameterized by the pose (translation and rotation) of the wrist joint relative to the PICO coordinate system. Additionally, the hand aperture (palm opening and closing) is mapped to the continuous/discrete state of the robot gripper. To facilitate a unified action representation, we formulate actions as the state deltas between consecutive frames, expressed in the reference frame of the preceding time step.

*Table 12.* Dataset Sampling Ratios for the Human Data Learning.

| Dataset | Ratio | Dataset | Ratio | Dataset | Ratio |
|---|---|---|---|---|---|
| Fractal | 0.117016 | Kuka | 0.117045 | Bridge | 0.122016 |
| Taco Play | 0.027175 | Jaco Play | 0.004455 | Berkeley Cable Routing | 0.002419 |
| Roboturk | 0.021427 | Viola | 0.008722 | Berkeley Autolab UR5 | 0.011192 |
| Toto | 0.018610 | Stanford Hydra Dataset | 0.040937 | Austin Buds Dataset | 0.001949 |
| NYU Franka Play Dataset | 0.007692 | Furniture Bench Dataset | 0.022558 | UCSD Kitchen Dataset | 0.000454 |
| Austin Sailor Dataset | 0.020175 | Austin Sirius Dataset | 0.015995 | DLR EDAN Shared Control | 0.000510 |
| IAMLab CMU Pickup Insert | 0.008356 | UTAustin Mutex | 0.020677 | Berkeley Fanuc Manipulation | 0.007155 |
| CMU Stretch | 0.001429 | BC-Z | 0.068743 | FMB Dataset | 0.019323 |
| DobbE | 0.013026 | DROID | 0.092715 | Human Manip. Data | 0.167972 |

**Training Strategy.** We explain the specific implementation of the two methods in section 3.4 in our experiment.

For the *w. human v.* variant: WoG is pretrained on the Open X-Embodiment (OXE) dataset for 100k training steps with a global batch size of 1024. At the second stage, all 1,920 hours of unannotated human videos are used with the OXE dataset. For robot data, the model receives supervision from both condition prediction and action prediction. For human video data, it receives supervision only from condition prediction. 50k steps of the same batch size is also used in the second stage.

For *w. human v./a.* variant: action supervision is applied to the 220-hour annotated subset in both training stages. Specifically,

in the first stage, the model utilizes both the OXE dataset and the 220-hour annotated human subset to perform action prediction conditioned on future observations. In the second stage, the training set is expanded to include the full 1,920-hour corpus of human manipulation videos alongside the OXE dataset. During this phase, action supervision is applied exclusively to the OXE data and the 220-hour annotated human subset, while the supervision for future condition prediction is applied to all data samples.

The dataset sampling ratios incorporated human dataset are shown in Table 12. We also show our human manipulation video samples in Figure 6.

### C.5. UMI Data Learning Settings

Our UMI data collection consists exclusively of egocentric observations, where actions are defined as egocentric motions relative to the headset coordinate system. Since our objective is to evaluate the performance improvement on our target tasks, we incorporate UMI data solely as a data source during the final fine-tuning stage alongside expert demonstrations. Consistent with the sampling strategy described earlier, it is assigned a sampling weight equivalent to that of the other expert demonstration tasks.

