# OpenReview forum: "World Guidance: World Modeling in Condition Space for Action Generation"
_ICML.cc/2026/Conference — ICML 2026 regular_

### Official Review · Reviewer_LWkX · 2026-03-07

**Soundness:** 3
**Presentation:** 3
**Significance:** 3
**Originality:** 3
**Overall Recommendation:** 4
**Confidence:** 4

**Summary:**

This paper proposes WoG (World Guidance), a framework that models future observations in a compact condition space tailored for action generation via a two-stage training pipeline. Extensive experiments across simulation (SIMPLER) and real-world robotic setups demonstrate that WoG outperforms state-of-the-art baselines in both action generation and generalization under out-of-distribution scenarios like background and lighting changes.

**Compliance With Llm Reviewing Policy:**

Affirmed.

**Final Justification:**

My concerns have been adequately addressed. I recommend the authors to add a brief discussion in the revised manuscript regarding the visual encoders abalation study.

**Key Questions For Authors:**

1. For the SIMPLER simulation experiments, only two vision encoder combinations (DINOv2+SigLIP, DINOv2+Wan VAE) are tested—how would WoG perform with lightweight visual encoders (e.g., ResNet)?

2. In the real-world OOD setup, lighting change is identified as the most challenging shift, but no qualitative analysis (e.g., failure cases, model behavior) is provided.

3. The UMI data experiment only reports overall success rate gains for P&P and towel folding tasks—are there any task-specific limitations to integrating UMI data, and how does the egocentric observation of UMI affect the condition space’s consistency with the robot’s third-person observation?

**Limitations:**

yes

**Strengths And Weaknesses:**

Strengths

1. Proposes a task-specific condition space for future world modeling, which inherently eliminates task-agnostic redundancy in visual representations.
2.  Enables scalable and flexible multi-source data utilization with two tailored strategies for human manipulation videos (annotated/unannotated) and UMI data, where unlabeled human videos only supervise condition prediction (no action annotation required) and UMI data is seamlessly integrated in fine-tuning.

Weaknesses

1. Limited performance on high spatial-constraint tasks: WoG shows only marginal gains on tasks requiring precise geometric/spatial reasoning (e.g., stacking blocks, drawer operation), as the condition space prioritizes dynamic motion over fine-grained spatial geometry modeling.

2.  Lack of ablations on condition space design: Critical design choices of the condition space (e.g., dimensionality, query token number in Q-Former) are not ablated or analyzed, leaving unclear how these hyperparameters affect model expressiveness and generalization.

3. No evaluation on longer action prediction horizons: The study only evaluates short-term action prediction (16 steps); there is no assessment of WoG’s performance and stability when predicting longer action sequences, a key requirement for real-world complex robotic manipulation.

---

> ### Author Rebuttal · Authors · 2026-03-31
>
> We thank the reviewer for acknowledging our contributions and for the constructive suggestions. We provide additional evaluations below to more comprehensively validate our method.
>
> > 💬 **W2**: *Ablations on condition space design.*
>
> A: Taking WoG with DINOv2 as the encoder, we ablate the condition space design. We compare:
> (1) Vanilla VLA without condition modeling or prediction;
> (2) a DreamVLA-style [1] variant, in which 256 tokens are used to query the last-layer VLM embeddings and align them with the full DINOv2 features;
> (3) WoG with 64, 16, and 8 query tokens in the Future Encoder, where the corresponding number of tokens are used to query the last-layer VLM embeddings and align them with the learned conditions.
> The SimplerEnv results are shown below.
>
> | Model | Visual Matching | Variant Aggregation | Overall Avg. |
> |-------|---------------------:|-------------------------:|--------------:|
> Vanilla VLA | 68.4% | 50.3% | 59.4% |
> | WoG  (Full DINO prediction) - 256 align tokens | 71.9% | 53.7% | 62.8% |
> | WoG (DINO) - 64 query tokens | 77.3% | 56.5% | 66.9% |
> | WoG (DINO) - 16 query tokens | **79.3%** | **59.6%** | **69.5%** |
> | WoG (DINO) - 8 query tokens | 73.6% | 58.3% | 66.0% |
>
> We find that aligning the full semantic space yields limited improvement over the baseline. With 64 tokens (a higher-dimensional condition space), WoG performs better on Visual Matching tasks but generalizes less well under Variant Aggregation. Conversely, the more compact 8-token representation shows stronger generalization but lower success rates on Visual Matching. This confirms a trade-off between expressiveness and generalization. The 16-token setting strikes a better balance, achieving strong performance in both task execution and generalization.
>
> > 💬 **W3**: *Longer action horizons.*
>
> A: We report results with the action horizon extended from 16 to 32 on SimplerEnv as shown below. Although success rates decrease slightly, the model maintains competitive performance overall.
> | Model | Visual Matching | Variant Aggregation | Overall Avg. |
> |-------|---------------------:|-------------------------:|--------------:|
> | WoG (DINO) Step 16 → 32 | 79.3% → 74.5% | 59.6% → 56.3% | 69.5% → 65.4% |
>
> > 💬 **Q1**: *Lightweight visual encoders.*
>
> A:   We compare WoG using a lightweight 80M ViT encoder (trained from scratch in Stage I) against the default pretrained 300M DINOv2 encoder. Results are shown below. WoG achieves strong performance even with the lightweight unpretrained encoder, though the pretrained model better adapts to scene variations in real2sim evaluation, yielding more robust performance.
> | Model | Visual Matching | Variant Aggregation | Overall Avg. |
> |-------|---------------------:|-------------------------:|--------------:|
> | WoG  (Lightweight ViT) | 74.5% | 56.6% | 65.6% |
> | WoG (DINO) | **79.3%** | **59.6%** | **69.5%** |
>
> > 💬 **Q2**: *Lighting change qualitative analysis.*
>
> A:  The challenge of lighting change lies in the prominent shadows cast on manipulated objects, a condition rarely captured in existing open-source datasets, leaving models poorly equipped to handle it. In the Fold task, shadows cause VLAs to mislocalize towel corners for grasping, where WoG still maintains a lead. In the P&P task, the cup casts a large shadow on the table, causing severe grasping deviations for all methods: over 20 trials, WoG achieves only 5% success while VPP and UniVLA both score 0%. We consider these results not statistically meaningful and thus did not report them in the main text. We will add a corresponding discussion in the revised version.
>
> > 💬 **Q3**: *UMI data experiment.*
>
> A: The UMI experiment verifies whether the model acquires viewpoint- and embodiment-agnostic condition modeling during pretraining. Unlike large-scale PICO data that can be incorporated during pretraining. UMI provides only a small amount of task-specific data at the fine-tuning stage, so we focus on its ability to enrich condition acquisition and boost success rates. Since WoG already achieves 100% on Close the Microwave, the UMI evaluation covers only P&P and Fold.
>
> **References**
> [1] Zhang et al. DreamVLA: A VLA Model Dreamed with Comprehensive World Knowledge. NeurIPS, 2025.

---

> > ### Author Rebuttal · Reviewer_LWkX · 2026-04-04
> >
> > My concerns have been adequately addressed. I recommend the authors to add a brief discussion in the revised manuscript regarding the visual encoders abalation study.

---

> > > ### Author Response · Authors · 2026-04-04
> > >
> > > Dear Reviewer LWkX:
> > >
> > > Thank you for the positive assessment of our work and our rebuttal. In the revised version, we will include the above visual encoder ablation study and add a brief discussion accordingly.
> > >
> > > Best Regards,
> > >
> > > Authors

---

### Official Review · Reviewer_ZVcr · 2026-03-12

**Soundness:** 3
**Presentation:** 2
**Significance:** 2
**Originality:** 3
**Overall Recommendation:** 5
**Confidence:** 3

**Summary:**

This paper introduces World Guidance (WoG), a framework for Vision-Language-Action (VLA) models that learn a compact condition space from future observations to guide the VLA model to generate precise actions. Experimental results demonstrate that WoG significantly enhances fine-grained robotic manipulation.

**Compliance With Llm Reviewing Policy:**

Affirmed.

**Final Justification:**

I thank the authors for their detailed response. My concerns have been fully addressed, and I have raised my score accordingly.

**Key Questions For Authors:**

1. Since there is a significant difference between the first-person observation images in the human data and the observation images from the robotic manipulator, is it necessary to establish a coordinate mapping between these two types of observations?

2. What is UMI data? Please clarify the difference between the annotated human data and UMI data.

3. How does the method handle undesirable future states when training with human demonstrations?

**Limitations:**

The authors should give an explict discussion on the limitations of the WoG framework. For example, show some falure cases and discuss the possible reasons behind them.

**Strengths And Weaknesses:**

Strengths:

1. Future-state–guided planning.

Leveraging future states to guide action planning is consistent with a result/value-oriented behavior generation paradigm. Furthermore, the second training stage internalizes the future-state prediction capability into the VLA model, leading to a relatively concise and efficient overall framework.

2. Comprehensive evaluation.

The paper evaluates the proposed method across three types of manipulation tasks—rigid-body manipulation, articulated-object manipulation, and deformable-object manipulation—which provides a relatively comprehensive validation of the accuracy of the generated actions.

3. Use of human data.

The method is capable of leveraging human demonstration data for training, and the paper provides experimental evidence supporting the effectiveness of this setting.

Concerns:

1. Insufficient description of the condition space construction.

The construction of the condition space is a key component of the proposed method. However, the process of computing $O_c$​ is not clearly described. The paper only states that it is “a learnable Q-Former–based encoder which queries action-relevant features and projects them into low-dimensional conditioning representations.” It would be helpful to provide a more explicit mathematical formulation of this process in the main text and to clearly annotate the corresponding component in Figure 2.

2. Causality vs. spurious correlations in learning $O_c$​.

A core challenge of the proposed approach is enabling the model to extract condition states $O_c$​ from future videos that are genuinely relevant to the task objective. Under the current training strategy, it is unclear how the method ensures that $O_c$​ captures causally relevant states rather than spurious correlations. It would be helpful to include an additional control experiment in the ablation study (Table 4). For example, under the WoG w/o cotrain setting, future video frames could be provided at test time to compute $O_c$​, and the resulting action generation accuracy could be compared with the baseline. Under this setup, the performance might even exceed that of the two-stage trained WoG, which would help demonstrate the effectiveness of the $O_c$​ extraction network after the first-stage training stage.

3. Potential negative-value future states in human data.

When training with human data, the future observation state $O_c$​ may not always correspond to a desirable outcome. In practice, human operators may realize during execution that the current state is not what they intended and subsequently correct it. Does the training dataset contain such cases? If so, can the model distinguish undesirable states and avoid generating actions that lead to them?

4. Discussion of related work.

The idea of guiding diffusion-based policies using future states or goal states has been explored in prior work, particularly in navigation and driving tasks. It would strengthen the paper to include a more detailed discussion of related work, for example:

A. Sridhar et al., “NoMaD: Goal Masking Diffusion Policies for Navigation and Exploration,” ICRA 2024.

---

> ### Author Rebuttal · Authors · 2026-03-31
>
> We thank the reviewer for the valuable feedback and the inspiring experimental suggestions. We address each concern below with additional discussions and experiments.
>
> > 💬 **W1**: *Insufficient description of the condition space construction.*
>
> A: The detailed architecture and mechanism of the Q-Former-based Future Encoder are provided in Appendix A (Figure 5). We will move them into the main text.
>
> > 💬 **W2**: *Additional experiments to test causality.*
>
> A: We fine-tune both the standard WoG and the Stage-I-only variant (which retains $O^c$ as auxiliary condition) on 95% of the real-world data, and evaluate normalized action MSE on the 5% validation set with ground-truth future frames. For the Stage-I-only variant, we test two modes: inference with and without $O^c$. Results are shown below.
>
> #### MSE error on validation set
>
> | Method | P&P ↓ | Fold ↓ | Close ↓ |
> |---|---:|---:|---:|
> | Stage I w/o Future Imgs | 0.043 | 0.058 | 0.018 |
> | Stage I w. Future Imgs | **0.007** | **0.018** | **0.005** |
> | Standard WoG | 0.011 | 0.022 | 0.006 |
>
> The variant without $O^c$ exhibits substantially higher prediction error, while introducing $O^c$ yields significantly improved accuracy, confirming the critical role of condition encoding for action prediction. The standard WoG with co-training, though slightly less accurate than the Stage-I model with ground-truth $O^c$, closely approaches its performance, validating the necessity of Stage II learning.
>
> > 💬 **W3 & Q3**: *Potential negative-value future states in human data.*
>
> A: Human actions, like robot action data, contain patterns that do not correspond to desirable outcomes. We use generative Action Head (DiT) to model this distribution, just as the Action Head also models suboptimal robot action distributions. Through this process, the VLM learns general understanding capabilities aligned with the dominant modes of the distribution. Similarly, the Future Encoder learns conditions that conform to common manipulation paradigms, guiding the model to identify generalizable conditions from the scene.
>
> > 💬 **W4**: *Discussion of related work.*
>
> A: We will include a discussion of future-guidance works represented by NoMaD in the revised version to better clarify the distinctiveness of our condition modeling approach.
>
> > 💬 **Q1 & Q2**: *Observation difference and data difference.*
>
> A: The OXE pretraining dataset already includes egocentric views (e.g., Google Robot) similar to human observations, and it also cover diverse third-person viewpoints. Since VLAs must learn unified behaviors across heterogeneous viewpoints, the condition space should generalize across perspectives as well, so no explicit coordinate transformation is applied.
>
> UMI [1] uses a hand-held gripper whose collected actions are transferable to a robot frame. PICO-based human data captures human wrist states in the PICO system and cannot be directly transferred, which is the key distinction. We will add this point in the main text.
>
> **References**
> [1] Chi et al. Universal manipulation
> interface: In-the-wild robot teaching without in-the-wild
> robots. In Proceedings of Robotics: Science and Systems
> (RSS), 2024.

---

> > ### Author Rebuttal · Reviewer_ZVcr · 2026-04-06
> >
> > My concerns have been fully addressed.  I recommend the authors to add these additional experiments and discussions in the revised manuscript.

---

> > > ### Author Response · Authors · 2026-04-06
> > >
> > > Dear Reviewer ZVcr:
> > >
> > > Thank you for your suggestions and the valuable feedback. We will add the experiments and discussions in the revised version.
> > >
> > > Best Regards,
> > >
> > > Authors

---

### Official Review · Reviewer_h17s · 2026-03-13

**Soundness:** 3
**Presentation:** 4
**Significance:** 3
**Originality:** 3
**Overall Recommendation:** 4
**Confidence:** 3

**Summary:**

This paper proposes WoG, a framework for robot action inference using world modeling that can generate guidance to support fine-grained action generation by learning compressed conditions associated with actions from future observations.
In the first stage of WoG, high-level representations extracted by a pretrained vision model are processed using a learnable Q-former-based encoder to query action-relevant features, which are then provided to the action head as guidance, enabling action prediction conditioned on both current and future information.
In the second stage, action prediction and future condition prediction are jointly learned, and by aligning the queried representations with the future conditional representations generated by the encoder, the knowledge of future conditions is transferred to the VLM backbone.
The performance of WoG was evaluated through benchmark simulation experiments in simpler environments and real-world experiments, and its superiority was validated through comparisons with existing world action model and latent action model methods.

**Compliance With Llm Reviewing Policy:**

Affirmed.

**Final Justification:**

This paper is generally well written and easy to follow, and the authors have conducted extensive experiments thoroughly to substantiate the superiority of the proposed framework.
Regarding the concern that there was a lack of qualitative comparison between representations learned with WoG and those learned without it, the authors have sufficiently addressed this issue by additionally presenting experimental results along with visualizations of the representations obtained with WoG.
Accordingly, I found the paper’s claims more convincing, and therefore I raised both the soundness score and my overall recommendation.

**Key Questions For Authors:**

1. What qualitative differences exist between the compressed conditional representations for future prediction obtained through WoG and those that are not compressed in this manner? For example, the representations obtained through WoG may be distributed with less redundancy across tasks in the representation space compared with those that are not. A direct comparative analysis of these representations could make the paper’s results more convincing.

2. How would the model’s performance change if WoG used only Vanilla VLA with Stage 1, or if the pretrained vision model features were fed directly into the action head without Stage 1? Such ablation results on WoG could facilitate a better understanding of the role of each stage in the framework.

**Limitations:**

yes

**Strengths And Weaknesses:**

### Strengths

- Extensive experiments:
This paper thoroughly validates the superiority of the proposed framework through extensive experiments, including not only experiments on various tasks in benchmark simulations but also real-world experiments, as well as comprehensive comparative evaluations against existing conventional VLA methods, world action models, and latent action models.

- Clear writing & representation:
Overall, the paper is clearly written and easy to follow, which helps readers understand the proposed framework and results. In addition, it effectively identifies the current status and limitations of related studies and clearly presents the objectives that this paper aims to address.

- Scalability:
The proposed framework allows the pretrained vision encoder to be replaced according to the objective, and its scalability is sufficiently established and validated through experiments in OOD environments and experiments leveraging large-scale human video data.

- Originality:
WoG further advances the ideas of prior studies that have used future predictive signals as guidance when VLA models generate actions by proposing a method for extracting a compressed future conditional representation for such guidance.
Moreover, by formulating this process and validating it through extensive experiments, it provides a more robust theoretical foundation than existing approaches regarding which future predictive signals should be utilized for action generation in VLA models.

### Weaknesses

- Lack of qualitative analysis:
In the introduction, the paper points out as a limitation that representations obtained by existing methods either contain substantial redundancy in a task-agnostic semantic space or provide only coarse guidance, making it difficult to support precise action generation. However, the paper appears to lack sufficient analysis of how the condition representations compressed by WoG concretely differ from those obtained by existing methods. Although the paper quantitatively validates the effectiveness of WoG based on success rates in manipulation tasks through benchmark simulations and real-world experiments, it does not provide direct comparisons, such as visualizing and comparing the internal embeddings of compressed and uncompressed conditions. As a result, it remains somewhat unclear whether such performance improvements are indeed attributable to the compact representations learned by WoG, as claimed in the paper.

- Lack of ablation experiments:
WoG consists of Vanilla VLA and the training stages of Stage 1 and Stage 2. Although the paper conducts an ablation study on the joint training in Stage 2, it does not report the experimental results for the case where only Vanilla VLA and Stage 1 are combined.

---

> ### Author Rebuttal · Authors · 2026-03-31
>
> We thank the reviewer for the positive assessment and the constructive suggestions for additional evaluation. Below we present supplementary experiments.
>
> > 💬 **W1 & Q1**: *Qualitative analysis of condition representations.*
>
> A: Using WoG with DINOv2 as the encoder, we evaluate the impact of condition representation. We compare:
> (1) Vanilla VLA without condition modeling or prediction;
> (2) a DreamVLA-style [1] variant, in which 256 tokens are used to query the last-layer VLM embeddings and align them with the full DINOv2 features;
> (3) WoG with 64, 16, and 8 query tokens in the Future Encoder, where the corresponding number of tokens are used to query the last-layer VLM embeddings and align them with the learned conditions.
> The SimplerEnv results are shown below.
>
> | Model | Visual Matching | Variant Aggregation | Overall Avg. |
> |-------|---------------------:|-------------------------:|--------------:|
> Vanilla VLA | 68.4% | 50.3% | 59.4% |
> | WoG  (Full DINO prediction) - 256 align tokens | 71.9% | 53.7% | 62.8% |
> | WoG (DINO) - 64 query tokens | 77.3% | 56.5% | 66.9% |
> | WoG (DINO) - 16 query tokens | **79.3%** | **59.6%** | **69.5%** |
> | WoG (DINO) - 8 query tokens | 73.6% | 58.3% | 66.0% |
>
> Predicting conditions consistently outperforms predicting full semantics regardless of the number of condition tokens, confirming that condition modeling captures more action-relevant representations while discarding redundant visual features.
>
> Among WoG variants, 64 tokens provide higher expressiveness but lower compactness, yielding stronger Visual Matching performance at the cost of weaker generalization under Variant Aggregation; 8 tokens exhibit the opposite pattern. This reveals a clear expressiveness-generalization trade-off. The 16-token setting achieves a better balance on both fronts, indicating that this configuration models and utilizes conditions more effectively.
>
> Meanwhile, we provide a t-SNE visualization of the condition embeddings and full semantic embeddings for 1,600 samples at [anonymous link](https://anonymous.4open.science/api/repo/imgs-8F28/file/comparasion.pdf?v=9eb62614).
>
> > 💬 **W2 & Q2**: *Lack of ablation experiments on stage I.*
>
> A: The Vanilla VLA + Stage I combination was not included because Stage I inference requires the future observation condition $O^c$, which is unavailable at deployment. As Reviewer **ZVcr** suggested in **Concern 2**, we evaluate this combination on a 5% held-out validation set of real-world data with ground-truth future frames.
>
> #### MSE error on validation set
>
> | Method | P&P ↓ | Fold ↓ | Close ↓ |
> |---|---:|---:|---:|
> | Stage I w/o Future Imgs | 0.043 | 0.058 | 0.018 |
> | Stage I w. Future Imgs | **0.007** | **0.018** | **0.005** |
> | Standard WoG | 0.011 | 0.022 | 0.006 |
>
> As shown in Table. We test two modes: with and without $O^c$. Introducing $O^c$ substantially reduces action MSE, confirming that the model learns highly useful conditions from future observations. After the standard WoG co-training in Stage II, the MSE converges close to the level achieved with ground-truth future frames, validating the effectiveness of Stage II for condition prediction.
>
> **References**
> [1] Zhang et al. DreamVLA: A VLA Model Dreamed with Comprehensive World Knowledge. NeurIPS, 2025.

---

> > ### Author Rebuttal · Reviewer_h17s · 2026-04-04
> >
> > Regarding the answer to W1, Q1:
> > The authors have shown in their response that WoG consistently outperforms approaches that predict the full semantics regardless of the number of tokens. In addition, by providing t-SNE visualizations for varying numbers of query tokens to qualitatively inspect the representations generated by WoG, they further strengthen the paper’s claims by demonstrating that WoG acquires more compact and effective representations.
> >
> > Regarding the answer to W2, Q2:
> > The authors’ argument is reasonable that the combination of Vanilla VLA and Stage 1 was not included in the comparison, since it is not available at deployment time.
> > Furthermore, by presenting a comparison table for the settings with and without future condition, the authors show that Stage 1 allows the model to learn useful conditioning, and that in Stage 2, knowledge of future conditioning can be successfully transferred to the VLM backbone.

---

> > > ### Author Response · Authors · 2026-04-04
> > >
> > > Dear Reviewer h17s:
> > >
> > > We sincerely thank you for the review and the positive feedback on our rebuttal. We will incorporate the comparision and visualizations into the revised version.
> > >
> > > Best Regards,
> > >
> > > Authors

---

### Official Review · Reviewer_MqAY · 2026-03-13

**Soundness:** 3
**Presentation:** 3
**Significance:** 3
**Originality:** 3
**Overall Recommendation:** 4
**Confidence:** 3

**Summary:**

The paper proposes WoG, a two-stage VLA framework that maps future observations into a compact condition space for guiding precise actions. Stage I injects learned conditions into the action head while stage II makes the VLM predict them internally. Experiments in simulation and the real world demonstrate that WoG significantly outperforms existing works.

**Compliance With Llm Reviewing Policy:**

Affirmed.

**Final Justification:**

Based on the rebuttal, I find the novelty clearer. However, as noted in my review, I am not an expert in this subfield and defer the final assessment of novelty to the AC. Therefore, I will keep my original score of weak accept.

**Key Questions For Authors:**

Overall, the paper reads well and is easy to understand. The writing is clear, the motivation is logical, and the experiments provide solid empirical support. My main concern is about novelty, as detailed in the weaknesses section. However, as I am not an expert in this subfield, I am not fully confident in this assessment and would like to see how other reviewers evaluate the novelty.

**Limitations:**

No, there is no dedicated Limitations section, and no discussion whatsoever of potential negative societal impacts

**Strengths And Weaknesses:**

**Strengths:**

- Clear motivation and framing of the trade-off between World Action Models (redundant rich features) and Latent Action Models (too coarse).

- Two-stage curriculum is logical: stage I discovers a useful bottleneck via supervision from future frames; stage II distils that bottleneck into the main VLM.

- Real-world results (Table 3) show meaningful gains over strong baselines (UniVLA, VPP) on articulated and deformable tasks and better retention under OOD shifts (background / novel object / lighting).

- Human-video and UMI integration experiments (Tables 5, Figure 4) are among the more interesting ablation suites, showing that even unlabelled human video can help when used only for condition prediction, and that UMI data gives large gains despite large domain gaps, supports the claim of capturing embodiment-agnostic dynamics.



**Weaknesses:**

I think the novelty is incremental. Based on my quick research, I think the core mechanism Stage I (using future observations as privileged input to find a task-relevant bottleneck), and Stage II (decoupling that input and requiring the VLA to predict it via an auxiliary head) has conceptual similarity with recent 2024–2025 works on auxiliary future prediction and trajectory-conditioned VLAs. I am not expert in this field, so I may be missing key distinctions.

---

> ### Author Rebuttal · Authors · 2026-03-25
>
> We thank the reviewer for the encouraging feedback and the insightful question on novelty. The concern is well-taken, and we clarify the key distinctions below.
>
> > 💬 **W1 & Q1**: *The novelty is incremental. The core mechanism has conceptual similarity with recent 2024–2025 works on auxiliary future prediction and trajectory-conditioned VLAs.*
>
> A:  Existing auxiliary prediction approaches—whether utilizing or predicting trajectories [1,2], flow [3], depth [4], or full features from foundation models [4,5,6]—share a common paradigm: they select a specific modality and predict its **full content as an auxiliary signal**, based on a modality-specific assumption (e.g., flow captures motion, depth aids spatial reasoning). The **key insight of WoG** is that predicting the full content of any modality provides only a subset of **action-relevant information** while introducing substantial redundancy, and is therefore **not a necessary condition** for action generation; meanwhile, predicting a single modality such as object trajectories may also fail to provide sufficient conditions for complete action inference. In other words, neither approach strikes the right balance between sufficiency and compactness.
>
> Rather than manually assuming which modality is useful, WoG constructs a pipeline that directly incorporates **future observations** into the action inference process and **allows the pipeline itself to discover the sufficient condition space**. This is precisely the objective of Stage I, and to the best of our knowledge, no prior work has explored this perspective.
>
> Stage II follows as a natural consequence: since future observations are unavailable at deployment, the model must learn to predict these conditions autonomously. While the use of an auxiliary prediction objective is shared with prior methods, the critical difference is that **WoG predicts a learned, task-driven condition space rather than predefined explicit features**—which is precisely why it achieves stronger performance and generalization.
>
> **References**
> [1] Zheng et al. TraceVLA: Visual Trace Prompting Enhances Spatial-Temporal Awareness for Generalist Robotic Policies. arXiv:2412.10345, 2024.
> [2] Su et al. Motion Before Action: Diffusing Object Motion as Manipulation Condition. IEEE RA-L, 2025.
> [3] Xu et al. Flow as the Cross-domain Manipulation Interface. CoRL, 2024.
> [4] Zhang et al. DreamVLA: A VLA Model Dreamed with Comprehensive World Knowledge. NeurIPS, 2025.
> [5] Ye et al. World Action Models are Zero-shot Policies. arXiv:2602.15922, 2026.
> [6] Lyu et al. LDA-1B: Scaling Latent Dynamics Action Model via Universal Embodied Data Ingestion. arXiv:2602.12215, 2026.

---

> > ### Author Rebuttal · Reviewer_MqAY · 2026-04-01
> >
> > Based on the rebuttal, I think the novelty is now more clearer but as I noted in the review I am not an expert in this subfield and would like to see how other reviewers evaluate the novelty.

---

> > > ### Author Response · Authors · 2026-04-04
> > >
> > > Dear Reviewer MqAY:
> > >
> > > We sincerely thank you for the review and the valuable feedback. We will clarify above points more explicitly in the revised version.
> > >
> > > Best Regards,
> > >
> > > Authors

---

### Decision · Program_Chairs · 2026-04-30

**Decision:**

Accept (regular)

**Comment:**

This paper presents World Guidance (WoG), a vision-language-action model that utilizes latent abstractions of future trajectories to generate actions. The primary insight of the paper lies in exploring the characterization of an effective latent space that could provide guidance in action predictions. The key idea is to define such a space through future predictions using pre-trained foundation models. Experiments are conducted on the SIMPLER simulation environment using two robotic configurations, demonstrating strong empirical results against state-of-the-art approaches.

The paper received overall positive reviews. Reviewers generally appreciated the clear motivation for the approach, the architectural design, and the strength of the empirical results. However, there were concerns raised on various fronts, including incremental novelty, the lack of sufficient analysis of how WoG’s condition representations differ from existing methods, and limitations in experiments. Authors provided a strong rebuttal addressing all the reviewers’ concerns, including additional empirical results.

AC had an independent reading of the paper and agrees with the reviewers that the paper makes a good technical contribution with strong empirical results. As such, AC recommends acceptance. Authors are encouraged to revise the paper to clarify the concerns raised by the reviewers, particularly regarding the condition space, spurious correlations, and to incorporate the new results reported during the rebuttal into the paper.